# Realistic Evaluation of Deep Semi-Supervised Learning Algorithms

**Avital Oliver**,* **Augustus Odena**,* **Colin Raffel**,* **Ekin D. Cubuk & Ian J. Goodfellow**
Google Brain
`{avitalo,augustusodena,craffel,cubuk,goodfellow}@google.com`

## Abstract

Semi-supervised learning (SSL) provides a powerful framework for leveraging unlabeled data when labels are limited or expensive to obtain. SSL algorithms based on deep neural networks have recently proven successful on standard benchmark tasks. However, we argue that these benchmarks fail to address many issues that SSL algorithms would face in real-world applications. After creating a unified reimplementation of various widely-used SSL techniques, we test them in a suite of experiments designed to address these issues. We find that the performance of simple baselines which do not use unlabeled data is often underreported, SSL methods differ in sensitivity to the amount of labeled and unlabeled data, and performance can degrade substantially when the unlabeled dataset contains out-of-distribution examples. To help guide SSL research towards real-world applicability, we make our unified reimplemention and evaluation platform publicly available.[2]

## 1 Introduction

It has repeatedly been shown that deep neural networks can achieve human- or super-human-level performance on certain supervised learning problems by leveraging large collections of labeled data. However, these successes come at a cost: Creating these large datasets typically requires a great deal of human effort (to manually label examples), pain and/or risk (for medical datasets involving invasive tests) or financial expense (to hire labelers or build the infrastructure needed for domain-specific data collection). For many practical problems and applications, we lack the resources to create a sufficiently large labeled dataset, which limits the wide-spread adoption of deep learning techniques.

An attractive approach towards addressing the lack of data is **semi-supervised learning** (SSL) [6]. In contrast with supervised learning algorithms, which require labels for all examples, SSL algorithms can improve their performance by also using unlabeled examples. SSL algorithms generally provide a way of learning about the structure of the data from the unlabeled examples, alleviating the need for labels. Some recent results [32, 50, 39] have shown that in certain cases, SSL approaches the performance of purely supervised learning, even when a substantial portion of the labels in a given dataset has been discarded. These results are demonstrated by taking an existing classification dataset (typically CIFAR-10 [31] or SVHN [40]) and only using a small portion of it as labeled data with the rest treated as unlabeled. The accuracy of a model trained with SSL with labeled and unlabeled data is then compared to that of a model trained on only the small labeled portion.

These recent successes raise a natural question: Are SSL approaches applicable in "real-world" settings? In this paper, we argue that this de facto way of evaluating SSL techniques does not address this question in a satisfying way. Our goal is to more directly answer this question by proposing a new

experimental methodology which we believe better measures applicability to real-world problems. Some of our findings include:

- When given equal budget for tuning hyperparameters, the gap in performance between using SSL and using only labeled data is smaller than typically reported.

- Further, a large classifier with carefully chosen regularization trained on a small labeled dataset with no unlabeled data can reach very good accuracy. This demonstrates the importance of evaluating different SSL algorithms on the same underlying model.

- In some settings, pre-training a classifier on a different labeled dataset and then retraining on only labeled data from the dataset of interest can outperform all SSL algorithms we studied.

- Performance of SSL techniques can degrade drastically when the unlabeled data contains a different distribution of classes than the labeled data.

- Different approaches exhibit substantially different levels of sensitivity to the amount of labeled and unlabeled data.

- Realistically small validation sets would preclude reliable comparison of different methods, models, and hyperparameter settings.

Separately, as with many areas of machine learning, direct comparison of approaches is confounded by the fact that minor changes to hyperparameters, model structure, training, etc. can have an outsized impact on results. To mitigate this problem, we provide a unified and modular software reimplementation of various state-of-the-art SSL approaches that includes our proposed evaluation techniques.

The remainder of this paper is structured as follows: In section 2, we enumerate the ways in which our proposed methodology improves over standard practice. In section 3, we give an overview of modern SSL approaches for deep architectures, emphasizing those that we include in our study. Following this discussion, we carry out extensive experiments (section 4) to better study the real-world applicability of our own reimplementation of various SSL algorithms. We restrict our analysis to image classification tasks as this is the most common domain for benchmarking deep learning models. Finally, we conclude (section 5) with concrete recommendations for evaluating SSL techniques.

## 2   Improved Evaluation

In this work, we make several improvements to the conventional experimental procedures used to evaluate SSL methods, which typically proceed as follows: First, take a common (typically image classification) dataset used for supervised learning and throw away the labels for most of the dataset. Then, treat the portion of the dataset whose labels were retained as a small labeled dataset $\mathcal{D}$ and the remainder as an auxiliary unlabeled dataset $\mathcal{D}_{UL}$. Some (not necessarily standard) model is then trained and accuracy is reported using the unmodified test set. The choice of dataset and number of retained labels is somewhat standardized across different papers. Below, we enumerate ways that we believe this procedure can be made more applicable to real-world settings.

**P.1  A Shared Implementation.** We introduce a shared implementation of the underlying architectures used to compare all of the SSL methods. This is an improvement relative to prior work because though the datasets used across different studies have largely become standardized over time, other experimental details vary significantly. In some cases, different reimplementations of a simple 13-layer convolutional network are used [32, 39, 50], which results in variability in some implementation details (parameter initialization, data preprocessing, data augmentation, regularization, etc.). Further, the training procedure (optimizer, number of training steps, learning rate decay schedule, etc.) is not standardized. These differences prevent direct comparison between approaches. All in all, these issues are not unique to SSL studies; they reflect a larger reproducibility crisis in machine learning research [28, 23, 13, 35, 38].

**P.2  High-Quality Supervised Baseline.** The goal of SSL is to obtain better performance using the combination of $\mathcal{D}$ and $\mathcal{D}_{UL}$ than what would be obtained with $\mathcal{D}$ alone. A natural baseline to compare against is the same underlying model (with modified hyperparameters) trained in a fully-supervised manner using only $\mathcal{D}$. While frequently reported, this baseline is occasionally omitted. Moreover, it is not always apparent whether the best-case performance has been eked out of the fully-supervised model (e.g. Laine & Aila [32] and Tarvainen & Valpola [50] both report a fully supervised baseline

*with ostensibly the same model* but obtain accuracies that differ between the two papers by up to 15%). To ensure that our supervised baseline is high-quality, we spent 1000 trials of hyperparameter optimization to tune both our baseline as well as all the SSL methods.

**P.3 Comparison to Transfer Learning.** In practice a common way to deal with limited data is to "transfer" a model trained on a separate, but similar, large labeled dataset [12, 51, 9]. This is typically achieved by initializing the parameters of a new model with those from the original model, and "fine-tuning" this new model using the small dataset. While this approach is only feasible when an applicable source dataset is available, it nevertheless provides a powerful, widely-used, and rarely reported baseline to compare against.

**P.4 Considering Class Distribution Mismatch.** Note that when taking an existing fully-labeled dataset and discarding labels, all members of $\mathcal{D}_{UL}$ come from the same classes as those in $\mathcal{D}$. In contrast, consider the following example: Say you are trying to train a model to distinguish between ten different faces, but you only have a few images for each of these ten faces. As a result, you augment your dataset with a large unlabeled dataset of images of random people's faces. In this case, it is extremely unlikely that any of the images in $\mathcal{D}_{UL}$ will be one of the ten people the model is trained to classify. Standard evaluation of SSL algorithms neglects to consider this possibility. This issue was indirectly addressed e.g. in [32], where labeled data from CIFAR-10 (a natural image classification dataset with ten classes) was augmented with unlabeled data from Tiny Images (a huge collection of images scraped from the internet). It also shares some characteristics with the related field of "domain adaptation", where the data distribution for test samples differs from the training distribution [4, 16]. We explicitly study the effect of differing class distributions between labeled and unlabeled data.

**P.5 Varying the Amount of Labeled *and* Unlabeled Data.** A somewhat common practice is to vary the size of $\mathcal{D}$ by throwing away different amounts of the underlyling labeled dataset [48, 43, 45, 50]. Less common is to vary the size of $\mathcal{D}_{UL}$ in a systematic way, which could simulate two realistic scenarios: First, that the unlabeled dataset is gigantic (e.g. using billions of unlabeled natural images on the Internet to augment a natural image classification task); or second, that the unlabeled dataset is also relatively small (e.g. in medical imaging, where both obtaining and labeling data is expensive).

**P.6 Realistically Small Validation Sets.** An unusual artefact of the way artificial SSL datasets are created is that often the validation set (data used for tuning hyperparameters and not model parameters) is significantly larger than the training set. For example, the standard SVHN [40] dataset has a validation set of roughly 7,000 labeled examples. Many papers that evaluate SSL methods on SVHN use only 1,000 labels from the training dataset but retain the full validation set. The validation set is thus over seven times bigger than the training set. Of course, in real-world applications, this large validation set would instead be used as the training set. The issue with this approach is that any objective values (e.g. accuracy) used for hyperparameter tuning would be significantly noisier across runs due to the smaller sample size from a realistically small validation set. In these settings, extensive hyperparameter tuning may be somewhat futile due to an excessively small collection of held-out data to measure performance on. In many cases, even using cross-validation may be insufficient and additionally incurs a substantial computational cost. The fact that small validation sets constrain the ability to select models is discussed in [6] and [14]. We take this a step further and directly analyze the relationship between the validation set size and variance in estimates of a model's accuracy.

## 3   Semi-Supervised Learning Methods

In supervised learning, we are given a training dataset of input-target pairs $(x, y) \in \mathcal{D}$ sampled from an unknown joint distribution $p(x, y)$. Our goal is to produce a prediction function $f_\theta(x)$ parametrized by $\theta$ which produces the correct target $y$ for previously unseen samples from $p(x)$. For example, choosing $\theta$ might amount to optimizing a loss function which reflects the extent to which $f_\theta(x) = y$ for $(x, y) \in \mathcal{D}$. In SSL we are additionally given a collection of unlabeled input datapoints $x \in \mathcal{D}_{UL}$, sampled from $p(x)$. We hope to leverage the data from $\mathcal{D}_{UL}$ to produce a prediction function which is more accurate than what would have been obtained by using $\mathcal{D}$ on its own.

From a broad perspective, the goal of SSL is to use $\mathcal{D}_{UL}$ to augment $f_\theta(x)$ with information about the structure of $p(x)$. For example, $\mathcal{D}_{UL}$ can provide hints about the shape of the data "manifold" which can produce a better estimate of the decision boundary between different possible target values.

A depiction of this concept on a simple toy problem is shown in fig. 1, where the scarcity of labeled data makes the decision boundary between two classes ambiguous but the additional unlabeled data reveals clear structure which can be discovered by an effective SSL algorithm.

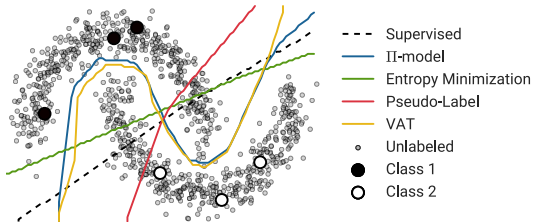

Figure 1: Behavior of the SSL approaches described in section 3 on the "two moons" dataset. We omit "Mean Teacher" and "Temporal Ensembling" (appendix A.1.2) because they behave like Π-Model (appendix A.1.1). Each approach was applied to a MLP with three hidden layers, each with 10 ReLU units. When trained on only the labeled data (large black and white dots), the decision boundary (dashed line) does not follow the contours of the data "manifold", as indicated by additional unlabeled data (small grey dots). In a simplified view, the goal of SSL is to leverage the unlabeled data to produce a decision boundary which better reflects the data's underlying structure.

A comprehensive overview of SSL methods is out of the scope of this paper; we refer interested readers to [53, 6]. Instead, we focus on the class of methods which solely involve adding an additional loss term to the training of a neural network, and otherwise leave the training and model unchanged from what would be used in the fully-supervised setting. We limit our focus to these approaches for the pragmatic reasons that they are simple to describe and implement and that they are currently the state-of-the-art for SSL on image classification datasets. Overall, the methods we consider fall into two classes: Consistency regularization, which enforces that realistic perturbations of data points $x \in \mathcal{D}_{UL}$ should not significantly change the output of $f_\theta(x)$; and entropy minimization, which encourages more confident predictions on unlabeled data. We now describe these methods in broad terms. See appendix A for more detail, and additional references to other SSL methods.

**Π-Model:** The simplest setting in which to apply consistency regularization is when the prediction function $f_\theta(x)$ is itself stochastic, i.e. it can produce different outputs for the same input $x$. This is quite common in practice during training when $f_\theta(x)$ is a neural network due to common regularization techniques such as data augmentation, dropout, and adding noise. Π-Model [32, 46] adds a loss term which encourages the distance between a network's output for different passes of $x \in \mathcal{D}_{UL}$ through the network to be small.

**Mean Teacher:** A difficulty with the Π-model approach is that it relies on a potentially unstable "target" prediction, namely the second stochastic network prediction which can rapidly change over the course of training. As a result, [50] proposed to obtain a more stable target output $\bar{f}_\theta(x)$ for $x \in \mathcal{D}_{UL}$ by setting the target to predictions made using an exponential moving average of parameters from previous training steps.

**Virtual Adversarial Training:** Instead of relying on the built-in stochasticity of $f_\theta(x)$, Virtual Adversarial Training (VAT) [39] directly approximates a tiny perturbation $r_{adv}$ to add to $x$ which would most significantly affect the output of the prediction function.

**Entropy Minimization (EntMin):** EntMin [21] adds a loss term applied that encourages the network to make "confident" (low-entropy) predictions for all unlabeled examples, regardless of their class.

**Pseudo-Labeling:** Pseudo-labeling [34] proceeds by producing "pseudo-labels" for $\mathcal{D}_{UL}$ using the prediction function itself over the course of training. Pseudo-labels which have a corresponding class probability which is larger than a predefined threshold are used as targets for a standard supervised loss function applied to $\mathcal{D}_{UL}$.

## 4 Experiments

In this section we cover issues with the evaluation of SSL techniques. We first address **P.1** and create a unified reimplementation of the methods outlined in section 3 using a common model architecture and training procedure. Our goal is *not* to produce state-of-the-art results, but instead to provide a rigorous comparative analysis in a common framework. Further, because our model architecture and training hyperparameters differ from those used to test SSL methods in the past, our results are not directly comparable to past work and should therefore be considered in isolation (see appendix D for

Table 1: Test error rates obtained by various SSL approaches on the standard benchmarks of CIFAR-10 with all but 4,000 labels removed and SVHN with all but 1,000 labels removed, using our proposed unified reimplementation. "Supervised" refers to using only 4,000 and 1,000 labeled datapoints from CIFAR-10 and SVHN respectively without any unlabeled data. VAT and EntMin refers to Virtual Adversarial Training and Entropy Minimization respectively (see section 3).

| Dataset | # Labels | Supervised | Π-Model | Mean Teacher | VAT | VAT + EntMin | Pseudo-Label |
|---------|----------|------------|---------|--------------|-----|--------------|--------------|
| CIFAR-10 | 4000 | $20.26 \pm .38\%$ | $16.37 \pm .63\%$ | $15.87 \pm .28\%$ | $13.86 \pm .27\%$ | $13.13 \pm .39\%$ | $17.78 \pm .57\%$ |
| SVHN | 1000 | $12.83 \pm .47\%$ | $7.19 \pm .27\%$ | $5.65 \pm .47\%$ | $5.63 \pm .20\%$ | $5.35 \pm .19\%$ | $7.62 \pm .29\%$ |

a full comparison). We use our reimplementation as a consistent testbed on which we carry out a series of experiments, each of which individually focusses on a single issue from section 2.

## 4.1 Reproduction

For our reimplementation, we selected a standard model that is modern, widely used, and would be a reasonable choice for a practitioner working on image classification. This avoids the possibility of using an architecture which is custom-tailored to work well with one particular SSL technique. We chose a Wide ResNet [52], due to their widespread adoption and availability. Specifically, we used "WRN-28-2", i.e. ResNet with depth 28 and width 2, including batch normalization [25] and leaky ReLU nonlinearities [36]. We did not deviate from the standard specification for WRN-28-2 so we refer to [52] for model specifics. For training, we chose the ubiquitous Adam optimizer [29]. For all datasets, we followed standard procedures for regularization, data augmentation, and preprocessing; details are in appendix B.

Given the model, we implemented each of the SSL approaches in section 3. To ensure that all of the techniques we are studying are given fair and equal treatment and that we are reporting the best-case performance under our model, we carried out a large-scale hyperparameter optimization. For every SSL technique in addition to a "fully-supervised" (not utilizing unlabeled data) baseline, we ran 1000 trials of Gaussian Process-based black box optimization using Google Cloud ML Engine's hyperparameter tuning service [18]. We optimized over hyperparameters specific to each SSL algorithm in addition to those shared across approaches.

We tested each SSL approach on the widely-reported image classification benchmarks of SVHN [40] with all but 1000 labels discarded and CIFAR-10 [31] with all but 4,000 labels discarded. This leaves 41,000 and 64,932 unlabeled images for CIFAR-10 and SVHN respectively when using standard validation set sizes (see appendix B). We optimized hyperparameters to minimize classification error on the standard validation set from each dataset, as is common practice (an approach we evaluate critically in section 4.6). Black-box hyperparameter optimization can produce unintuitive hyperparameter settings which vary unnecessarily between different datasets and SSL techniques. We therefore audited the best solutions found for each dataset/SSL approach combination and hand-designed a simpler, unified set of hyperparameters. Hyperparameter values were selected if they were shared across different SSL approaches and achieved comparable performance to those found by the tuning service. After unification, the only hyperparameters which varied across different SSL algorithms were the learning rate, consistency coefficient, and any hyperparameters unique to a given algorithm (e.g. VAT's $\epsilon$ hyperparameter). An enumeration of these hyperparameter settings can be found in appendix C.

We report the test error at the point of lowest validation error for the hyperparameter settings we chose in table 1. We use this hyperparameter setting without modification in all of our experiments.

## 4.2 Fully-Supervised Baselines

By using the same budget of hyperparameter optimization trials for our fully-supervised baselines, we believe we have successfully addressed item **P.2**. For comparison, table 2 shows the fully-supervised baseline and SSL error rates listed in prior studies. **We find the gap between the fully-supervised baseline and those obtained with SSL is smaller in our study than what is generally reported in the literature.** For example, [32] report a fully-supervised baseline error rate of 34.85% on CIFAR-10 with 4000 labels which is improved to 12.36% using SSL; our improvement for the same approach went from 20.26% (fully-supervised) to 16.37% (with SSL).

Table 2: Reported change in error rate from fully-supervised (no unlabeled data) to SSL. We do not report results for VAT because a fully-supervised baseline was not reported in [39]. We also did not include the SVHN results from [46] because they use 732 labeled examples instead of 1000.

| Method | CIFAR-10 4000 Labels | SVHN 1000 Labels |
|---|---|---|
| Π-Model [32] | $34.85\% \rightarrow 12.36\%$ | $19.30\% \rightarrow 4.80\%$ |
| Π-Model [46] | $13.60\% \rightarrow 11.29\%$ | – |
| Π-Model (ours) | $20.26\% \rightarrow 16.37\%$ | $12.83\% \rightarrow 7.19\%$ |
| Mean Teacher [50] | $20.66\% \rightarrow 12.31\%$ | $12.32\% \rightarrow 3.95\%$ |
| Mean Teacher (ours) | $20.26\% \rightarrow 15.87\%$ | $12.83\% \rightarrow 5.65\%$ |

Table 3: Comparison of error rate using SSL and transfer learning. VAT with Entropy Minimization was the most performant method on CIFAR-10 in our experiments. "No overlap" refers to transferring from an ImageNet model which was not trained on classes which are similar to the classes in CIFAR-10 (see section 4.3 for details).

| Method | CIFAR-10 4000 Labels |
|---|---|
| VAT with Entropy Minimization | 13.13% |
| ImageNet → CIFAR-10 | 12.09% |
| ImageNet → CIFAR-10 (no overlap) | 12.91% |

We can push this line of enquiry even further: Can we design a model with a regularization, data augmentation, and training scheme which can match the performance of SSL techniques without using any unlabeled data? Of course, comparing the performance of this model to SSL approaches with different models is unfair; however, we want to understand the upper-bound of fully-supervised performance as a benchmark for future work.

After extensive experimentation, we chose the large Shake-Shake model of [17] due to its powerful regularization capabilities. We used a standard data-augmentation scheme consisting of random horizontal flips and random crops after zero-padding by 4 pixels on each side [22], as well as cutout regularization with a patch length of 16 pixels [11]. Training and regularization was as in [17], with a learning rate of 0.025 and weight decay factor of 0.0025. On 4,000 labeled data points from CIFAR-10, this model obtained an average test error of **13.4%** over 5 independent runs. This result emphasizes the importance of the underlying model in the evaluation of SSL algorithms, and reinforces our point that **different algorithms must be evaluated using the same model to avoid conflating comparison.**

## 4.3 Transfer Learning

Further to the point of item **P.2**, we also studied the technique of transfer learning using a pre-trained classifier, which is frequently used in limited-data settings but often neglected in SSL studies. We trained our standard WRN-28-2 model on ImageNet [10] downsampled to 32x32 [7] (the native image size of CIFAR-10). We used the same training hyperparameters as used for the supervised baselines reported in section 4.1. Then, we fine-tuned the model using 4,000 labeled data points from CIFAR-10. As shown in table 3, the resulting model obtained an error rate of **12.09%** on the test set. **This is a lower error rate than any SSL technique achieved using this network, indicating that transfer learning may be a preferable alternative when a labeled dataset suitable for transfer is available.** Note that we did not tune our model architecture or hyperparameters to improve this transfer learning result - we simply took the baseline model from our SSL experiments and used it for transfer learning. This suggests that the error rate of 12.09% is a conservative estimate of the potential performance of transfer learning in this setting.

Note that ImageNet and CIFAR-10 have many classes in common, suggesting that this result may reflect the best-case application of transfer learning. To test how much this overlap affected transfer learning performance, we repeated the experiment after removing the 252 ImageNet classes (listed in appendix F) which were similar to any of the CIFAR-10 classes. Performance degraded moderately to 12.91%, which is comparable to the best SSL technique we studied. We also experimented with transfer learning from ImageNet to SVHN, which reflects a much more challenging setting requiring substantial domain transfer. We were unable to achieve convincing results when transferring to SVHN which suggests that the success of transfer learning may heavily depend on how closely related the two datasets are. More concretely, it primarily demonstrates that the transfer learning on a separate, related, and labeled dataset can provide the network with a better learning signal than SSL can using unlabeled data. We are interested in exploring the combination of transfer learning and SSL in future work.

## 4.4 Class Distribution Mismatch

Now we examine the case where labeled and unlabeled data come from the same underlying distribution (e.g. natural images), but the unlabeled data contains classes not present in the labeled data. This setting violates the strict definition of semi-supervised learning given in section 3, but as outlined in item **P.4** it nevertheless represents a common use-case for SSL (for example, augmenting a face recognition dataset with unlabeled images of people not in the labeled set). To test this, we synthetically vary the class overlap in our common test setting of CIFAR-10. Specifically, we perform 6-class classification on CIFAR-10's animal classes (bird, cat, deer, dog, frog, horse). The unlabeled data comes from four classes — we vary how many of those four are among the six labeled classes to modulate class distribution mismatch. We also compare to a fully supervised model trained using no unlabeled data. As before, we use 400 labels per class for CIFAR-10, resulting in 2400 labeled examples.

Our results are shown in fig. 2. **We demonstrate the surprising result that adding unlabeled data from a mismatched set of classes can actually *hurt* performance compared to not using any unlabeled data at all** (points above the black dotted line in fig. 2). This implies that it may be preferable to pay a larger cost to obtain labeled data than to obtain unlabeled data if the unlabeled data is sufficiently unrelated to the core learning task. However, we did not re-tune hyperparameters these experiments; it is possible that by doing so the gap could narrow.

## 4.5 Varying Data Amounts

Many SSL techniques are tested only in the core settings we have studied so far, namely CIFAR-10 with 4,000 labels and SVHN with 1,000 labels. However, we argue that varying the amount of labeled data tests how performance degrades in the very-limited-label regime, and also at which point the approach can recover the performance of training with all of the labels in the dataset. We therefore ran experiments on both SVHN and CIFAR with different labeled data amounts; the results are shown in fig. 4. In general, the performance of all of the SSL techniques tends to converge as the number of labels grows. On SVHN, VAT exhibits impressively consistent performance across labeled data amounts, where in contrast the performance of $\Pi$-Model is increasingly poor as the number of labels decreases. As elsewhere, we emphasize that these results only apply to our specific architecture and hyperparameter settings and may not provide general insight into each algorithms' behavior.

Another possibility is to vary the amount of unlabeled data. However, using the CIFAR-10 and SVHN datasets in isolation places an upper limit on the amount of unlabeled data available. Fortunately, SVHN is distributed with the "SVHN-extra" dataset, which adds 531,131 additional digit images and which was previously used as unlabeled data in [50]. Similarly, the "Tiny Images" dataset can augment CIFAR-10 with eighty million additional unlabeled images as done in [32], however it also introduces a class distribution mismatch between labeled and unlabeled data because its images are not necessarily from the classes covered by CIFAR-10. As a result, we do not consider Tiny Images for auxiliary unlabeled data in this paper.

We evaluated the performance of each SSL technique on SVHN with 1,000 labels and varying amounts of unlabeled data from SVHN-extra, which resulted in the test errors shown in fig. 3. As expected, increasing the amount of unlabeled data tends to improve the performance of SSL techniques. However, we found that performance levelled off consistently across algorithms once 80,000 unlabeled examples were available. Furthermore, performance seems to degrade slightly for Pseudo-Labeling and $\Pi$-Model as the amount of unlabeled data increases. More broadly, **we find surprisingly different levels of sensitivity to varying data amounts across SSL techniques.**

## 4.6 Small Validation Sets

In all of the experiments above (and in recent experiments in the literature that we are aware of), hyperparameters are tuned on a labeled validation set which is significantly larger than the labeled portion of the training set. We measure the extent to which this provides SSL algorithms with an unrealistic advantage compared to real-world scenarios where the validation set would be smaller.

We can derive a theoretical estimate for the number of validation samples required to confidently differentiate between the performance of different approaches using Hoeffding's inequality [24]:

$$\mathbf{P}(|\bar{V} - \mathbb{E}[V]| < p) > 1 - 2\exp(-2np^2) \tag{1}$$

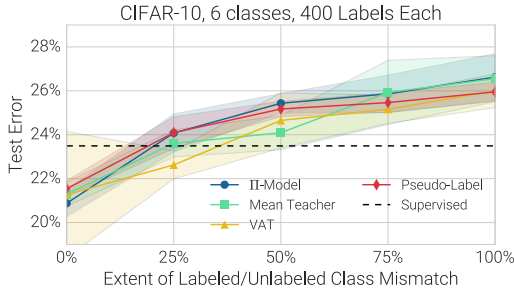

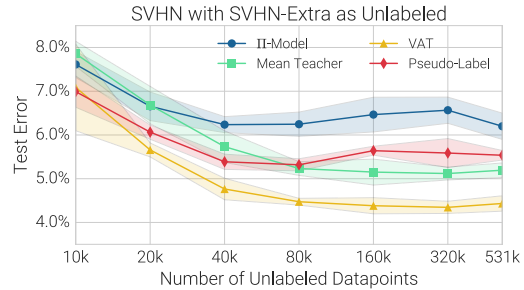

Figure 2: Test error for each SSL technique on CIFAR-10 (six animal classes) with varying overlap between classes in the labeled and unlabeled data. For example, in "25%", one of the four classes in the unlabeled data is not present in the labeled data. "Supervised" refers to using no unlabeled data. Shaded regions indicate standard deviation over five trials.

Figure 3: Test error for each SSL technique on SVHN with 1,000 labels and varying amounts of unlabeled images from SVHN-extra. Shaded regions indicate standard deviation over five trials. X-axis is shown on a logarithmic scale.

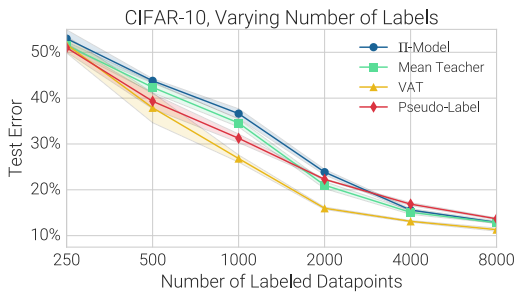

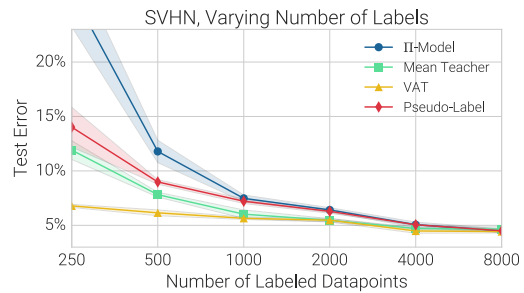

Figure 4: Test error for each SSL technique on SVHN and CIFAR-10 as the amount of labeled data varies. Shaded regions indicate standard deviation over five trials. X-axis is shown on a logarithmic scale.

where in our case $\bar{V}$ is the empirical estimate of the validation error, $\mathbb{E}[V]$ is its hypothetical true value, $p$ is the desired maximum deviation between our estimate and the true value, and $n$ is the number of examples in the validation set. In this analysis, we are treating validation error as the average of independent binary indicator variables denoting whether a given example in the validation set is classified correctly or not. As an example, if we want to be 95% confident that our estimate of the validation error differs by less than 1% absolute of the true value, we would need nearly 20,000 validation examples. This is a disheartening estimate due to the fact that the difference in test error achieved by different SSL algorithms reported in table 1 is often close to or smaller than 1%, but 20,000 is many times more samples than are provided in the training sets.

This theoretical analysis may be unrealistic due to the assumption that the validation accuracy is the average of independent variables. To measure this phenomenon empirically, we took baseline models trained with each SSL approach on SVHN with 1,000 labels and evaluated them on validation sets with varying sizes. These synthetic small validation sets were sampled randomly and without overlap from the full SVHN validation set. We show the mean and standard deviation of validation error over 10 randomly-sampled validation sets for each method in fig. 5. For validation sets of the same size (100%) as the training set, some differentiation between the approaches is possible. **However, for a realistically-sized validation set (10% of the training set size), differentiating between the performance of the models is not feasible.** This suggests that SSL methods which rely on heavy hyperparameter tuning on a large validation set may have limited real-world applicability. Cross-validation can help with this problem, but the reduction of variance may still be insufficient and its use would incur an N-fold computational increase.

A possible objection to this experiment is that there may be strong correlation between the accuracy of different SSL techniques when measured on the same validation set. If that is indeed the case, then principled model selection may be possible because all that is necessary is choosing the better of a class of models, not estimating each model's expected error as an exact number. In order to account for this objection, in fig. 6 we show the mean and standard deviation of the *difference in validation error* between each SSL model and Π-model (chosen arbitrarily as a point of comparison).

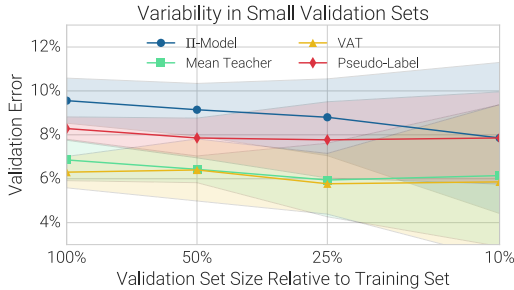
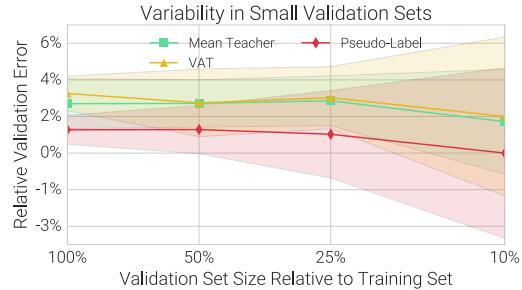

Figure 5: Average validation error over 10 randomly-sampled nonoverlapping validation sets of varying size. For each SSL approach, we re-evaluated an identical model on each randomly-sampled validation set. The mean and standard deviation of the validation error over the 10 sets are shown as lines and shaded regions respectively. Models were trained on SVHN with 1,000 labels. Validation set sizes are listed relative to the training size (e.g. 10% indicates a size-100 validation set). X-axis is shown on a logarithmic scale.

Figure 6: Average and standard deviation of relative error over 10 randomly-sampled nonoverlapping validation sets of varying size. The experimental set-up is identical to the one in fig. 5, with the following change: The mean and standard deviation are computed over the difference in validation error compared to Π-model, rather than the absolute validation error. X-axis is shown on a logarithmic scale.

For realistically small validation sets under this setting, the overlap between the error bounds with small validation sets still surpasses the difference between the error for different models. Thus, we still argue that with realistically small validation sets, model selection may not be feasible.

## 5 Conclusions and Recommendations

Our experiments provide strong evidence that standard evaluation practice for SSL is unrealistic. What changes to evaluation should be made to better reflect real-world applications? Our recommendations for evaluating SSL algorithms are as follows:

- Use the exact same underlying model when comparing SSL approaches. Differences in model structure or even implementation details can greatly impact results.

- Report well-tuned fully-supervised and transfer learning performance where applicable as baselines. The goal of SSL should be to significantly outperform the fully-supervised settings.

- Report results where the class distribution mismatch systematically varies. We showed that the SSL techniques we studied all suffered when the unlabeled data came from different classes than the labeled data — a realistic scenario that to our knowledge is drastically understudied.

- Vary both the amount of labeled and unlabeled data when reporting performance. An ideal SSL algorithm is effective even with very little labeled data and benefits from additional unlabeled data. Specifically, we recommend combining SVHN with SVHN-Extra to test performance in the large-unlabeled-data regime.

- Take care not to over-tweak hyperparameters on an unrealistically large validation set. A SSL method which requires significant tuning on a per-model or per-task basis in order to perform well will not be useable when validation sets are realistically small.

Our discoveries also hint towards settings where SSL is most likely the right choice for practitioners:

- When there are no high-quality labeled datasets from similar domains to use for fine-tuning.

- When the labeled data is collected by sampling i.i.d. from the pool of the unlabeled data, rather than coming from a (slightly) different distribution.

- When the labeled dataset is large enough to accurately estimate validation accuracy, which is necessary when doing model selection and tuning hyperparameters.

SSL has seen a great streak of successes recently. We hope that our results and publicly-available unified implementation help push these successes towards the real world.

## Acknowledgements

We thank Rodrigo Benenson, Andrew Dai, Sergio Guadarrama, Roy Frostig, Aren Jansen, Alex Kurakin, Katherine Lee, Roman Novak, Jon Shlens, Jascha Sohl-Dickstein, Jason Yosinski and many other members of the Google Brain team for feedback and fruitful discussions. We also thank our anonymous reviewers for their helpful comments on an early draft of our manuscript.

## Footnotes

[2]`https://github.com/brain-research/realistic-ssl-evaluation`

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
