[Supplementary Material · neurips2018_supplementary.pdf]

# A    Semi-supervised learning methods, in more detail

There have been a wide variety of proposed SSL methods, including "transductive" [15] variants of $k$-nearest neighbors [27] and support vector machines [26], graph-based methods [53, 5], and algorithms based on learning features (frequently via generative modeling) from unlabeled data [2, 33, 47, 8, 19, 30, 43, 41, 48].

A comprehensive overview is out of the scope of this paper; we instead refer interested readers to [53, 6].

We now describe the methods we analyze in this paper (as described in section 3) in more detail.

## A.1    Consistency Regularization

Consistency regularization describes a class of methods with following intuitive goal: Realistic perturbations $x \rightarrow \hat{x}$ of data points $x \in \mathcal{D}_{UL}$ should not significantly change the output of $f_\theta(x)$. Generally, this involves minimizing $d(f_\theta(x), f_\theta(\hat{x}))$ where $d(\cdot, \cdot)$ measures a distance between the prediction function's outputs, e.g. mean squared error or Kullback-Leibler divergence. Typically the gradient through this consistency term is only backpropagated through $f_\theta(\hat{x})$. In the toy example of fig. 1, this would ideally result in the classifier effectively separating the two class clusters due to the fact that their members are all close together. Consistency regularization can be seen as a way of leveraging the unlabeled data to find a smooth manifold on which the dataset lies [3]. This simple principle has produced a series of approaches which are currently state-of-the-art for SSL.

### A.1.1    Stochastic Perturbations/Π-Model

The simplest setting in which to apply consistency regularization is when the prediction function $f_\theta(x)$ is itself stochastic, i.e. it can produce different outputs for the same input $x$. This is common when $f_\theta(x)$ is a neural network due to common regularization techniques such as data augmentation, dropout, and adding noise. These regularization techniques themselves are typically designed in such a way that they ideally should not cause the model's prediction to change, and so are a natural fit for consistency regularization.

The straightforward application of consistency regularization is thus minimizing $d(f_\theta(x), f_\theta(\hat{x}))$ for $x \in \mathcal{D}_{UL}$ where in this case $d(\cdot, \cdot)$ is chosen to be mean squared error. This distance term is added to the classification loss as a regularizer, scaled by a weighting hyperparameter. This idea was first proposed in [1] and later studied in [46] and [32], and has been referred to as "Pseudo-Ensembles", "Regularization With Stochastic Transformations and Perturbations" and the "Π-Model" respectively. We adopt the latter name for its conciseness. In fig. 1, the Π-Model successfully finds the correct decision boundary.

### A.1.2    Temporal Ensembling/Mean Teacher

A difficulty with the Π-model approach is that it relies on a potentially unstable "target" prediction, namely the second stochastic network prediction which can rapidly change over the course of training. As a result, [50] and [32] proposed two methods for obtaining a more stable target output $\bar{f}_\theta(x)$ for $x \in \mathcal{D}_{UL}$. "Temporal Ensembling" [32] uses an exponentially accumulated average of outputs of $f_\theta(x)$ for the consistency target. Inspired by this approach, "Mean Teacher" [50] instead proposes to use a prediction function parametrized by an exponentially accumulated average of $\theta$ over training. As with the Π-model, the mean squared error $d(f_\theta(x), \bar{f}_\theta(x))$ is added as a regularization term with a weighting hyperparameter. In practice, it was found that the Mean Teacher approach outperformed Temporal Ensembling [50], so we will focus on it in our later experiments.

### A.1.3    Virtual Adversarial Training

Instead of relying on the built-in stochasticity of $f_\theta(x)$, Virtual Adversarial Training (VAT) [39] directly approximates a tiny perturbation $r_{adv}$ to add to $x$ which would most significantly affect the output of the prediction function. An approximation to this perturbation can be computed efficiently as

$$r \sim \mathcal{N}\left(0, \frac{\xi}{\sqrt{\dim(x)}} I\right) \tag{2}$$

$$g = \nabla_r d(f_\theta(x), f_\theta(x + r)) \tag{3}$$

$$r_{adv} = \epsilon \frac{g}{||g||} \tag{4}$$

where $\xi$ and $\epsilon$ are scalar hyperparameters. Consistency regularization is then applied to minimize $d(f_\theta(x), f_\theta(x + r_{adv}))$ with respect to $\theta$, effectively using the "clean" output as a target given an adversarially perturbed input. VAT is inspired by adversarial examples [49, 20], which are natural datapoints $x$ which have a virtually imperceptible perturbation added to them which causes a trained model to misclassify the datapoint. Like the $\Pi$-Model, the perturbations caused by VAT find the correct decision boundary in fig. 1.

### A.2   Entropy-Based

A simple loss term which can be applied to unlabeled data is to encourage the network to make "confident" (low-entropy) predictions for all examples, regardless of the actual class predicted. Assuming a categorical output space with $K$ possible classes (e.g. a $K$-dimensional $\mathrm{softmax}$ output), this gives rise to the "entropy minimization" term [21]:

$$-\sum_{k=1}^{K} f_\theta(x)_k \log f_\theta(x)_k \tag{5}$$

Ideally, entropy minimization will discourage the decision boundary from passing near data points where it would otherwise be forced to produce a low-confidence prediction [21]. However, given a high-capacity model, another valid low-entropy solution is simply to create a decision boundary which has overfit to locally avoid a small number of data points, which is what appears to have happened in the synthetic example of fig. 1 (see appendix E for further discussion). On its own, entropy minimization has not been shown to produce competitive results compared to the other methods described here [45]. However, entropy minimization was combined with VAT to obtain state-of-the-art results by [39]. An alternative approach which is applicable to multi-label classification was proposed by [45], but it performed similarly to entropy minimization on standard "one-hot" classification tasks. Interestingly, entropy *maximization* was also proposed as a regularization strategy for neural networks by [42].

### A.3   Pseudo-Labeling

Pseudo-labeling [34] is a simple heuristic which is widely used in practice, likely because of its simplicity and generality – all that it requires is that the model provides a probability value for each of the possible labels. It proceeds by producing "pseudo-labels" for $\mathcal{D}_{UL}$ using the prediction function itself over the course of training. Pseudo-labels which have a corresponding class probability which is larger than a predefined threshold are used as targets for a standard supervised loss function applied to $\mathcal{D}_{UL}$. While intuitive, it can nevertheless produce incorrect results when the prediction function produces unhelpful targets for $\mathcal{D}_{UL}$, as shown in fig. 1. Note that pseudo-labeling is quite similar to entropy regularization, in the sense that it encourages the model to produce higher-confidence (lower-entropy) predictions for data in $\mathcal{D}_{UL}$ [34]. However, it differs in that it only enforces this for data points which already had a low-entropy prediction due to the confidence thresholding. Pseudo-labeling is also closely related to self-training [44, 37], which differs only in the heuristics used to decide which pseudo-labels to retain. The Pseudo-labeling paper [34] also discusses using unsupervised pre-training; we did not implement this in our experiments.

# B Dataset details

Overall, we followed standard data normalization and augmentation practice. For SVHN, we converted image data to floating point values in the range [-1, 1]. For data augmentation, we solely used random translation by up to 2 pixels. We used the standard train/validation split, with 65,932 images for training and 7,325 for validation.

For any model which was to be used to classify CIFAR-10 (e.g. including the base ImageNet model for the transfer learning experiment in section 4.3), we applied global contrast normalization and ZCA-normalized the inputs using statistics calculated on the CIFAR-10 training set. ZCA normalization is a widely-used and surprisingly important preprocessing step for CIFAR-10. Data augmentation on CIFAR-10 included random horizontal flipping, random translation by up to 2 pixels, and Gaussian input noise with standard deviation 0.15. We used the standard train/validation split, with 45,000 images for training and 5,000 for validation.

Table 4: Hyperparameter settings used in our experiments. All hyperparameters were tuned via large-scale hyperparameter optimization and then distilled to sensible and unified defaults by hand. Adam's $\beta_1$, $\beta_2$, and $\epsilon$ parameters were left to the defaults suggested by [29]. *Following [50], we ramped up the consistency coefficient starting from 0 to its maximum value using a sigmoid schedule so that it achieved its maximum value at 200,000 iterations. **We found that CIFAR-10 and SVHN required different values for $\epsilon$ in VAT (6.0 and 1.0 respectively), likely due to the difference in how the input is normalized in each dataset.

| Shared | |
| --- | --- |
| L1 regularization coefficient | 0.001 |
| L2 regularization coefficient | 0.0001 |
| Learning decayed by a factor of | 0.2 |
| at training iteration | 400,000 |
| Consistency coefficient rampup* | 200,000 |
| **Supervised** | |
| Initial learning rate | 0.003 |
| **$\Pi$-Model** | |
| Initial learning rate | 0.0003 |
| Max consistency coefficient | 20 |
| **Mean Teacher** | |
| Initial learning rate | 0.0004 |
| Max consistency coefficient | 8 |
| Exponential moving average decay | 0.95 |
| **VAT** | |
| Initial learning rate | 0.003 |
| Max consistency coefficient | 0.3 |
| VAT $\epsilon$ | 6.0 or 1.0** |
| VAT $\xi$ | $10^{-6}$ |
| **VAT + EM** (as for VAT) | |
| Entropy penalty multiplier | 0.06 |
| **Pseudo-Label** | |
| Initial learning rate | 0.003 |
| Max consistency coefficient | 1.0 |
| Pseudo-label threshold | 0.95 |

## C  Hyperparameters

In our hyperparameter search, for each SSL method, we always separately optimized algorithm-agnostic hyperparameters such as the learning rate, its decay schedule and weight decay coefficients. In addition, we optimized to those hyperparameters specific to different SSL approaches separately for each approach. In keeping with our argument in section 4.6, we attempted to find hyperparameter settings which were performant across datasets and SSL approaches so that we could avoid unrealistic tweaking. After hand-tuning, we used the hyperparameter settings summarized in table 4, which lists those settings which were shared and common to all SSL approaches.

We trained all networks for 500,000 updates with a batch size of 100. We did not use any form of early stopping, but instead continuously monitored validation set performance and report test error at the point of lowest validation error. All models were trained with a single worker on a single GPU (i.e. no asynchronous training).

Table 5: Test error rates obtained by various SSL approaches on the standard benchmarks of CIFAR-10 with all but 4,000 labels removed and SVHN with all but 1,000 labels removed. Top: Reported results in the literature; Bottom: Using our proposed unified reimplementation. "Supervised" refers to using only 4,000 and 1,000 labeled datapoints from CIFAR-10 and SVHN respectively without any unlabeled data. VAT + EntMin refers Virtual Adversarial Training with Entropy Minimization (see section 3). Note that the model used for results in the bottom has roughly half as many parameters as most models in the top (see section 4.1).

| Method | CIFAR-10<br>4000 Labels | SVHN<br>1000 Labels |
|---|---|---|
| Π-Model [46] | 11.29% | – |
| Π-Model [32] | 12.36% | 4.82% |
| Mean Teacher [50] | 12.31% | 3.95% |
| VAT [39] | 11.36% | 5.42% |
| VAT + EntMin [39] | 10.55% | 3.86% |
| **Results above this line cannot be directly compared to those below** | | |
| Supervised | $20.26 \pm 0.38\%$ | $12.83 \pm 0.47\%$ |
| Π-Model | $16.37 \pm 0.63\%$ | $7.19 \pm 0.27\%$ |
| Mean Teacher | $15.87 \pm 0.28\%$ | $5.65 \pm 0.47\%$ |
| VAT | $13.86 \pm 0.27\%$ | $5.63 \pm 0.20\%$ |
| VAT + EntMin | $13.13 \pm 0.39\%$ | $5.35 \pm 0.19\%$ |
| Pseudo-Label | $17.78 \pm 0.57\%$ | $7.62 \pm 0.29\%$ |

# D    Comparison of our results with reported results in the literature

In table 5, we show how our results compare to what has been reported in the literature. Our numbers cannot be directly compared to those previously reported due to a lack of a shared underlying network architecture. For example, our model has roughly half as many parameters as the one used in [32, 39, 50], which may partially explain its somewhat worse performance. Our findings are generally consistent with these; namely, that all of these SSL methods improve (to a varying degree) over the baseline. Further, Virtual Adversarial Training and Mean Teacher both appear to work best, which is consistent with their shared state-of-the-art status.

# E  Decision boundaries found by Entropy Minimization cut through the unlabeled data

Why does Entropy Minimization not find good decision boundaries in the "two moons" figure (fig. 1)? Even though a decision boundary that avoids both clusters of unlabeled data would achieve low loss, so does any decision boundary that's extremely confident and "wiggles" around each individual unlabeled data point. The neural network easily overfits to such a decision boundary simply by increasing the magnitude of its output logits. Figure 7 shows how training changes the decision contours.

Figure 7: Predictions made by a model trained with Entropy Minimization, as made at initialization, and after 125 and 1000 training steps. Points where the model predicts "1" or "2" are shown in red or blue, respectively. Color saturation corresponds to prediction confidence, and the decision boundary is the white line. Notice that after 1000 steps of training the model is extremely confident at every point, which achieves close to zero prediction entropy on unlabeled points.

# F    Classes in ImageNet which overlap with CIFAR-10

Table 6: Classes in ImageNet which are similar to one of the classes in CIFAR-10. For reference, the CIFAR-10 classes are airplane, automobile, bird, cat, deer, dog, frog, horse, ship and truck.

| ID | Description | ID | Description | ID | Description | ID | Description |
|---|---|---|---|---|---|---|---|
| 7 | cock | 152 | Japanese spaniel | 215 | Brittany spaniel | 283 | Persian cat |
| 8 | hen | 153 | Maltese dog | 216 | clumber | 284 | Siamese cat |
| 9 | ostrich | 154 | Pekinese | 217 | English springer | 285 | Egyptian cat |
| 10 | brambling | 155 | Shih-Tzu | 218 | Welsh springer spaniel | 286 | cougar |
| 11 | goldfinch | 156 | Blenheim spaniel | 219 | cocker spaniel | 287 | lynx |
| 12 | house finch | 157 | papillon | 220 | Sussex spaniel | 288 | leopard |
| 13 | junco | 158 | toy terrier | 221 | Irish water spaniel | 289 | snow leopard |
| 14 | indigo bunting | 159 | Rhodesian ridgeback | 222 | kuvasz | 290 | jaguar |
| 15 | robin | 160 | Afghan hound | 223 | schipperke | 291 | lion |
| 16 | bulbul | 161 | basset | 224 | groenendael | 292 | tiger |
| 17 | jay | 162 | beagle | 225 | malinois | 293 | cheetah |
| 18 | magpie | 163 | bloodhound | 226 | briard | 403 | aircraft carrier |
| 19 | chickadee | 164 | bluetick | 227 | kelpie | 404 | airliner |
| 20 | water ouzel | 165 | black-and-tan coonhound | 228 | komondor | 405 | airship |
| 21 | kite | 166 | Walker hound | 229 | Old English sheepdog | 408 | amphibian |
| 22 | bald eagle | 167 | English foxhound | 230 | Shetland sheepdog | 436 | beach wagon |
| 23 | vulture | 168 | redbone | 231 | collie | 466 | bullet train |
| 24 | great grey owl | 169 | borzoi | 232 | Border collie | 468 | cab |
| 30 | bullfrog | 170 | Irish wolfhound | 233 | Bouvier des Flandres | 472 | canoe |
| 31 | tree frog | 171 | Italian greyhound | 234 | Rottweiler | 479 | car wheel |
| 32 | tailed frog | 172 | whippet | 235 | German shepherd | 484 | catamaran |
| 80 | black grouse | 173 | Ibizan hound | 236 | Doberman | 510 | container ship |
| 81 | ptarmigan | 174 | Norwegian elkhound | 237 | miniature pinscher | 511 | convertible |
| 82 | ruffed grouse | 175 | otterhound | 238 | Greater Swiss Mountain dog | 554 | fireboat |
| 83 | prairie chicken | 176 | Saluki | 239 | Bernese mountain dog | 555 | fire engine |
| 84 | peacock | 177 | Scottish deerhound | 240 | Appenzeller | 569 | garbage truck |
| 85 | quail | 178 | Weimaraner | 241 | EntleBucher | 573 | go-kart |
| 86 | partridge | 179 | Staffordshire bullterrier | 242 | boxer | 575 | golfcart |
| 87 | African grey | 180 | American Staffordshire terrier | 243 | bull mastiff | 581 | grille |
| 88 | macaw | 181 | Bedlington terrier | 244 | Tibetan mastiff | 586 | half track |
| 89 | sulphur-crested cockatoo | 182 | Border terrier | 245 | French bulldog | 595 | harvester |
| 90 | lorikeet | 183 | Kerry blue terrier | 246 | Great Dane | 609 | jeep |
| 91 | coucal | 184 | Irish terrier | 247 | Saint Bernard | 612 | jinrikisha |
| 92 | bee eater | 185 | Norfolk terrier | 248 | Eskimo dog | 625 | lifeboat |
| 93 | hornbill | 186 | Norwich terrier | 249 | malamute | 627 | limousine |
| 94 | hummingbird | 187 | Yorkshire terrier | 250 | Siberian husky | 628 | liner |
| 95 | jacamar | 188 | wire-haired fox terrier | 251 | dalmatian | 654 | minibus |
| 96 | toucan | 189 | Lakeland terrier | 252 | affenpinscher | 656 | minivan |
| 97 | drake | 190 | Sealyham terrier | 253 | basenji | 661 | Model T |
| 98 | red-breasted merganser | 191 | Airedale | 254 | pug | 675 | moving van |
| 99 | goose | 192 | cairn | 255 | Leonberg | 694 | paddlewheel |
| 100 | black swan | 193 | Australian terrier | 256 | Newfoundland | 705 | passenger car |
| 127 | white stork | 194 | Dandie Dinmont | 257 | Great Pyrenees | 717 | pickup |
| 128 | black stork | 195 | Boston bull | 258 | Samoyed | 724 | pirate |
| 129 | spoonbill | 196 | miniature schnauzer | 259 | Pomeranian | 734 | police van |
| 130 | flamingo | 197 | giant schnauzer | 260 | chow | 751 | racer |
| 131 | little blue heron | 198 | standard schnauzer | 261 | keeshond | 757 | recreational vehicle |
| 132 | American egret | 199 | Scotch terrier | 262 | Brabancon griffon | 779 | school bus |
| 133 | bittern | 200 | Tibetan terrier | 263 | Pembroke | 780 | schooner |
| 134 | crane | 201 | silky terrier | 264 | Cardigan | 803 | snowplow |
| 135 | limpkin | 202 | soft-coated wheaten terrier | 265 | toy poodle | 814 | speedboat |
| 136 | European gallinule | 203 | West Highland white terrier | 266 | miniature poodle | 817 | sports car |
| 137 | American coot | 204 | Lhasa | 267 | standard poodle | 829 | streetcar |
| 138 | bustard | 205 | flat-coated retriever | 268 | Mexican hairless | 833 | submarine |
| 139 | ruddy turnstone | 206 | curly-coated retriever | 269 | timber wolf | 847 | tank |
| 140 | red-backed sandpiper | 207 | golden retriever | 270 | white wolf | 864 | tow truck |
| 141 | redshank | 208 | Labrador retriever | 271 | red wolf | 867 | trailer truck |
| 142 | dowitcher | 209 | Chesapeake Bay retriever | 272 | coyote | 871 | trimaran |
| 143 | oystercatcher | 210 | German short-haired pointer | 273 | dingo | 874 | trolleybus |
| 144 | pelican | 211 | vizsla | 274 | dhole | 895 | warplane |
| 145 | king penguin | 212 | English setter | 275 | African hunting dog | 908 | wing |
| 146 | albatross | 213 | Irish setter | 281 | tabby | 913 | wreck |
| 151 | Chihuahua | 214 | Gordon setter | 282 | tiger cat | 914 | yawl |