[Reviews · NeurIPS 2018]

Reviewer 1



This paper proposes a systematic evaluation of SSL methods, studies the pitfalls of current approaches to evaluation, and, conducts experiments to show the impact of rigorous validation on kinds of conclusions we can draw from these methods. I really like the paper and read it when it appeared on arXiv back in April. In many places we are lacking these kind of systematic approaches to robust evaluations and it's refreshing to see more of these papers emerging that question the foundation of our validation methodologies and provide a coherent evaluation. Suggestions for improvements: - The paper mainly deals with two image categorisation datasets. While these methods have been studied in many recent SSL papers, they also have their own limitations, some of which is mentioned in the paper. But the main problem is that it restricts them to a single domain which is image categorisation. It would have been interesting if we could see how these methods behave on other domains (e.g. NLP tasks, Audio/Speech, ...). - In terms of SSL methods, it would have been interesting to study an instance of graph/manifold based methods (e.g. "Deep Learning via Semi-Supervised Embedding", Weston et al, ICML 2008) as that would've completed the spectrum of traditional assumptions/approaches to SSL (density/margin based models (EntMin/TSVM), self-training (pseudo-labelling), co-training(might not be as applicable as you don't have data with multiple views), manifold based models (LapSVM/SemiEmb)). - Following from this, I would actually suggest that self-training should be also one of the default baselines as it's so trivial to implement that it basically is a good baseline to have around for any task. While pseudo-labelling paper [34] seems like doing that I'm not sure if the implementation details from the paper were followed (e.g. only using pseudo-labelling as a fine-tuning step or using a denoising AE) or you really performed a more classical self-training. - In Sec 4.2 where you improve the performance of the supervised model by utilising [16, 21, 11] and obtain a performance that's comparable to those of SSL with the baseline method, while you acknowledge that it's unfair to make the comparison, but what is preventing you from actually applying SSL on this highly optimised model and see how far down the test error goes? In other words, do SSL methods provide model-invariant benefit or we will only see improvements if we just used the baseline model? - In Sec 4.3, would be interesting to see if you removed the CIFAR-10 classes from ImageNet and re-ran the experiments what you would get out? Does your SVHN conclusion in this section generalises to CIFAR-10? - In Sec 4.5, Although outside of the scope of the paper do you have any intuition why in Fig 4b the performance of VAT is actually very good across different number of labelled samples? - In Sec 4.6, I would suggest to actually do experiments by restricting the size of the validation set to as you say 10% or so of the training set (or perform cross-validation) and report performance on the test set based on hyper-parameter optimisation on the smaller dev set.

Reviewer 2



The paper proposes a novel framework to consistently evaluate semi-supervised learning (SSL) algorithms. The authors pose a number of interesting questions, e.g. what happens when labeled and unlabeled data come from different distributions, or what happens when only small validation sets are available. Experimental findings show that SSL evaluation might need some re-thinking, making the contribution of the paper significant. Once released, the evaluation platform will foster further research. Note that I'm not very familiar with the literature on SSL, so I'm not sure that the set of methods considered in Sec. 3 is complete. The clarity of the paper is satisfactory. Sentences at l. 218-219, pg. 4, could be made clearer (how was the set of hyperparameters hand-designed?). In some plots (Fig. 2, Fig. 5), shaded regions are not-so-well visible. A few observations/questions: - It would be interesting to discuss real-world scenarios in which the class distribution mismatch between labeled/unlabeled data is as high as the one considered in Sec. 4.4. - Do the authors think that the choices they made in terms of architecture/optimization technique for the platform can have any particular impact on the exposed results (is it expected that different architectures /algorithms would lead to the same conclusions / did the authors explore different choices?)? - The authors focus on image classification tasks. Do they have any insights on how their findings would apply to different tasks? Minor things: - Pg. 4, line 177, typo: applied

Reviewer 3



This paper evaluates popular semi-supervised learning (SSL) algorithms, namely, Pi-Model, Mean Teacher, Virtual Adversarial Training, Entropy Minimization and Pseudo-labelling in a unified manner. The authors point out several issues in the current practices for evaluating SSL algorithms, like not using the same budget for identifying hyper-parameters for the fully-supervised baseline, using an unrealistically large validation set to tune hyper-parameters, etc. Based on extensive numerical experiments on CIFAR-10 and SVHN datasets, the authors recommend best practices for evaluating SSL algorithms as well as suggest settings where SSL makes sense. Strengths: • The paper is well-written. The sections on summary of key findings and recommendations are useful to the SSL community. • The experiments on varying the amount of labeled and unlabeled data, class mismatch between labeled and unlabeled examples are a nice contribution. • The unified implementation used in this work is made available to the public. Comments: • At various places, key statements are highlighted in bold font, which is quite unusual and distracting. These statements are clearly summarized in the beginning as well as the end of the paper, so the use of bold sentences throughout the paper seems rather unnecessary. • Line 161: The line should read “for more details”. • In Section 4.2, the meaning of the phrase “averaged over 5 runs” is a little bit ambiguous. I am not sure which of the following is true: o The model is trained 5 times to calculate 5 error rates, and the average error rate is reported. If so, then how many models are trained for the results displayed in Table 1? o The predictions are averaged across 5 runs, and the error rate of the resulting ensemble of models is reported. If so, it is not fair to compare this with the results in Table 1, as ensembling will improve the performance of SSL algorithms as well. • A few numbers describing the supervised learning baseline results shown by others would help back up the statement that the baseline results are often underreported (similar to what is done in Table 3).